# Impact of Road Marking Retroreflectivity on Machine Vision in Dry Conditions: On-Road Test

**DOI:** 10.3390/s22041303

**Published:** 2022-02-09

**Authors:** Darko Babić, Dario Babić, Mario Fiolić, Arno Eichberger, Zoltan Ferenc Magosi

**Affiliations:** 1Faculty of Transport and Traffic Sciences, University of Zagreb, Vukelićeva 4, 10000 Zagreb, Croatia; darko.babic@fpz.unizg.hr (D.B.); mario.fiolic@fpz.unizg.hr (M.F.); 2Institute of Automotive Engineering, Graz University of Technology, Inffeldgasse 11/II, A-8010 Graz, Austria; arno.eichberger@tugraz.at (A.E.); zoltan.magosi@tugraz.at (Z.F.M.)

**Keywords:** ADAS, lane support systems, lane detection, automated driving, visibility, retroreflection

## Abstract

(1) Background: Due to its high safety potential, one of the most common ADAS technologies is the lane support system (LSS). The main purpose of LSS is to prevent road accidents caused by road departure or entrance in the lane of other vehicles. Such accidents are especially common on rural roads during nighttime. In order for LSS to function properly, road markings should be properly maintained and have an adequate level of visibility. During nighttime, the visibility of road markings is determined by their retroreflectivity. The aim of this study is to investigate how road markings’ retroreflectivity influences the detection quality and the view range of LSS. (2) Methods: An on-road investigation comprising measurements using Mobileye and a dynamic retroreflectometer was conducted on four rural roads in Croatia. (3) Results: The results show that, with the increase of markings’ retroreflection, the detection quality and the range of view of Mobileye increase. Additionally, it was determined that in “ideal” conditions, the minimal value of retroreflection for a minimum level 2 detection should be above 55 mcd/lx/m^2^ and 88 mcd/lx/m^2^ for the best detection quality (level 3). The results of this study are valuable to researchers, road authorities and policymakers.

## 1. Introduction

When looking at road accidents statistics, one can conclude that lane departure represents one of the most common causes of road accidents. Just in Croatia, around 45% of all accidents with fatalities and serious injuries can be related to some form of lane departure [1]. It is similar in other countries across the world. Namely, a recent US study highlighted that 51% of all fatal accidents are caused by lane departure [2]. A majority of such accidents occur on rural roads [3,4] due to higher driving speed, changing road geometry, distracted driving and/or fatigue as main influencing factors.

A vast body of literature is focused on solving the aforementioned problem, and a majority of such studies are focused on different road markings’ measures as a cost–benefit solution [5,6,7,8,9]. Although the investigated measures showed a significant reduction in lane departure crashes, the problem still exists, and the accidents statistics show slow improvements. A major leap forward could be achieved with different vehicle safety technologies which have been developed in the last decade. These technologies are divided into passive ones, which reduce injuries sustained by passengers when an accident occurs, and active ones who try to keep a vehicle under control and avoid accidents [10]. 

Lane support systems (LSS) are one of the main active systems designed to assist the driver in maintaining proper lateral movement of the vehicle, i.e., driving in his driving lane by means of sound, dashboard display reminder, steering wheel vibration and automatic steering force [11]. 

Current LSS use passive vision-based cameras and image processing to collect and analyze data from roads [12]. In general, lane detection starts with image pre-processing that includes different corrections of the collected image (such as exposure correction and shadow removal) and feature extraction. This is followed by feature detection, model fitting and time integration to keep temporal and position consistency [13,14].

Due to their high potential, LSS were the subject of investigations in several studies. A couple of studies predicted their potential in reducing lane departure crashes. Depending on the penetration rate, vehicle type and crash data, it is estimated that between 6% and 23% of fatal crashes could be prevented with LSS [11,15,16,17].

On the other hand, only a few studies evaluated the effectiveness of LSS on real-world crashes. One of the first such studies investigated the effect of lane departure warning on large trucks using carrier-collected data. Overall, the results show that trucks equipped with LSS had a 48% lower crash rate for the relevant crashes (single-vehicle run-off-road, head-on and sideswipe crashes) compared to trucks without LSS [18]. An analysis of the Swedish national statistics showed that LSS yielded a statistically significant reduction of 53%, with a lower limit of 11% for head-on and single-vehicle crashes on roads with higher speed limits (70–120 km/h) [19]. Two studies in 2018 resulted in similar conclusions. Spicer et al. [20] based their research on crashes which included BMW vehicles in a US fleet, and found that vehicles equipped with LSS and auto emergency braking were involved in between 13% and 63% fewer accidents (depending on vehicle type and crash type). Cicchino [21] concluded that, without accounting for driver demographics, vehicles with LSS had significantly lower crash rates of all severities (18%), in those with injuries (24%) and in those with fatalities (86%). However, the author highlights that the effect of LSS was lower when driver demographics were added as a control variable—an 11% reduction in crashes of all severities and a 21% reduction in crashes with injuries.

Although LSS clearly show a significant safety potential, their effectiveness is influenced by several factors [12,14,22,23]: camera quality (focal distance and camera velocity), condition, colour, width and visibility of lane markings (daytime visibility, night-time visibility—retroreflection and contrast to pavement), lane marking configuration (full/dashed, length of dashed lines), driving speed, weather conditions, general visibility of the environment, sun direction, pavement characteristics (type, condition and texture), road geometry, type of road edge (structured/unstructured) as well as combinations of the above factors. From all the above, the quality of markings, weather conditions, road geometry and general visibility of the environment may be the most important factors. Namely, Wilson et al. (2007) highlighted that LSS work properly only 36% of the time when road markings are not clear [24]. Additionally, Gordon et al. (2010) found that these systems can achieve 90% of their capacity during daytime but only 20% during nighttime because of light conditions [25]. After adjusting several factors, the overall effectiveness was estimated to range from 13% to 31%. A Swedish study also highlighted markings’ visibility and suggested that the luminance coefficient must be at least 5 mcd/lx/m^2^ higher than the road surface and that it should be at least 85 mcd/lx/m^2^ [26]. Potters Industry and Mobileye study (2016) concluded that the range of view of the investigated machine-vision is between 6–18 m in front of the vehicle and that the retroreflectivity of road markings affected the reading quality. Namely, markings with higher retroreflectivity increased the reading level and confidence [27]. A similar result was concluded in a US study [28]. The machine-vision detection of lane markings increased with the increase of retroreflection and contrast ratio. In general, such systems detect markings with a minimal retroreflectivity of 100 mcd/lx/m^2^ but do not necessarily provide the strongest detection. 

An extensive study was conducted in Australia with the aim of determining the implications of road markings for machine vision [12]. The study included several vehicles equipped with a Mobileye camera in order to test marking detection in different scenarios. These scenarios included the impact of different road marking characteristics (daytime dry luminance coefficient, daytime dry and wet contrast ratio, night dry and wet retroreflectivity, night dry and wet contrast ratio and marking width), different complex situations (such as road markings’ perceptual measures), non-marked edge lines, road curvation, etc. Based on the data analysis, it was concluded that lane detection during daytime was generally less effective than at nighttime due to the complexity of visual clutter evident during daylight hours and the fact that retroreflective properties of well-maintained lane markings provide greater contrast during nighttime. Additionally, it was highlighted that the contrast ratio between lane markings and the surrounding substrate should be between 5-to-1 and 10-to-1 during nighttime and 3-to-1 during daytime. 

The literature review leads us to the conclusion that the proper functioning of LSS is affected by a number of factors, one of which one is the visibility of road markings (daytime and nighttime), which is generally considered as their most important property [29]. Daytime visibility represents the quotient of the luminance of the field of a road marking in a given direction by the illuminance on the field. In contrast, nighttime visibility represents the quotient of the luminance L of the field of a road marking in the direction of observation by the illuminance E⊥ at the field perpendicular to the direction of the incident light and is achieved by the use of retroreflective materials (glass beads) [30]. Although the visibility of road markings is one of the most important factors which impact the LSS, there is still insufficient knowledge on the required visibility levels, especially in nighttime conditions, which would facilitate the proper functioning of LSS. Moreover, the studies that investigated this topic were mainly conducted on a testing track (except the Australian study [12]), where conditions are very different from real road conditions. 

Therefore, the aim of this paper is to investigate how nighttime visibility, i.e., the retroreflectivity of road markings, affects the functioning of LSS. For this purpose, an on-road investigation (real road conditions) was conducted using a dynamic retroreflectometer and a Mobileye camera. The obtained results may help in developing standards and minimal requirements for road markings and in planning and optimizing their maintenance activities, all with the aim of facilitating the proper functioning of ADAS and thus increasing road safety. The detailed methodology of the study is presented in the following chapters.

## 2. Methodology

### 2.1. Instruments

Nighttime visibility was measured using a dynamic retroreflectometer (Zehntner ZDR 6020) mounted on the testing vehicle of the Faculty of Transport and Traffic Sciences (University of Zagreb, Zagreb, Croatia). The used method enables constant measuring of retroreflectivity (R_L_) while driving by measuring the reflection of light rays from the tested surface at an angle of 2.29° with an entrance angle of 1.24° and at a distance of 30 m [30]. 

The device was calibrated prior to the testing according to the calibration procedure prescribed by the manufacturer. The measuring interval was set at 50 m, meaning that every 50 m, an average retroreflectivity value of the respective interval was calculated and recorded. Such settings were previously used in [31]. 

The data relating to lane detection was recorded using a Mobileye 630 system implemented in the testing vehicle (BMW640i). Using image processing chips, a Mobileye camera enables high-performance real-time image processing of different objects on roads such as lane markings, pedestrians, etc. For the purpose of this study, we recorded data related to the type of detected longitudinal marking (continuous or dashed), approximate marking width, view range and the quality of only middle marking. Technical specifications of the Mobileye system are presented in Table 1.

The vehicle was also equipped with a precise measurement system to record the vehicle’s trajectory by a combination of GPS localization (Novatel OEM-6–RT2 receiver) and inertial measurement unit (GENESYS ADMA G-III).

Measuring vehicles equipped with instruments are shown in Figure 1. 

### 2.2. Procedure

The study was conducted on four road sections in Croatia with a total length of 120.8 km. The roads in the majority of their length pass through a rural area and partially through settlements. Furthermore, these roads represent a typical two-way road with a lane width of 3.5 m and low traffic volumes. The roads were selected based on the fact that they are rural with low traffic volumes. Namely, rural roads with low traffic volumes have a higher risk of road departure accidents [2], and thus the importance of LSS is increased. Furthermore, rural roads are often less maintained, compared to motorways, for example, and thus the quality of road markings often significantly differs on different sections of the same road. Therefore, due to safety issues, changing geometry and different quality levels of road markings, for the purpose of this study, we identified rural roads as the “worst-case scenario” for LSS. Additionally, roads were selected if they had the recommended width of lane markings for LSS [32,33,34].

Since edge markings were not present on the whole length of the roads, the analysis is based only on middle lines. All markings were white, 15 cm wide and developed from solventborne paint (Type I).

The main characteristics of road sections are presented in Table 2.

Measurements with both instruments were conducted during one night (21 September 2020) in dry conditions and under a clear sky. Driving speed during the measurement was in accordance with the speed limit and varied between 60 km/h and 80 km/h. During the tests, two researchers were present in both vehicles presented in Figure 1—A driver and an operator who controlled the beginning and the end of each measurement. The researchers present in the vehicle for measuring road markings retroreflection were employees of an accredited laboratory for testing the quality of road markings and thus have extensive experience in conducting dynamic retroreflection measurements of road markings. Additionally, the researchers in the vehicle equipped with Mobileye conducted a number of tests using Mobileye and have excellent knowledge on how the system operates.

### 2.3. Data Analysis

The data from the dynamic retroreflectometer, as described in Section 2.1, was recorded on the basis of a 50 m interval which represented the average retroreflectivity (R_L_) of the respected interval. Additionally, retroreflectivity values were categorized into four groups: (1) <100 mcd/lx/m^2^; (2) ≥100 < 200 mcd/lx/m^2^; (3) ≥200 < 300 mcd/lx/m^2^; (4) ≥300 mcd/lx/m^2^.

With Mobileye, we recorded the view range and the detection quality of road markings. The raw data were extracted using a Control Area Network (CAN) bus interface separately for each road. The view range was determined in meters (maximal value 80 m), while the detection quality level was ranked on the scale from 0 to 3, where 0 equalled “nothing detected”, 1 presented “low detection confidence”, 2 “medium detection confidence” and 3 “high detection confidence”. The thresholds for each quality level are not known to the authors since this information is a “know-how” of the manufacturer. The sampling rate of the camera was set at 100 Hz. 

Since the aim of the study was to analyze how retroreflectivity (R_L_) alone affects the detection by LSS, smaller parts of the roads that pass through inhabited areas with road lighting were excluded from the analysis.

Using the QGIS tool, we overlapped the data, i.e., joined retroreflectivity values, to the Mobileye data. After that, average values of Mobileye’s quality readings and view range were calculated for each 50 m interval. 

The normal distribution of retroreflectivity (R_L_) and the view range were tested using the Kolmogorov–Smirnov test, while Spearman’s correlation coefficient was used to measure the association between retroreflectivity, range of view and detection quality. The Kruskal–Wallis test was used to identify a statistically significant difference in range of view between retroreflectivity categories as well as a significant difference in average retroreflectivity between detection quality categories. A more detailed analysis of the Kruskal–Wallis test was conducted with a series of Mann–Whitney U tests with a Bonferroni-corrected level of statistical significance. Finally, the Receiver Operating Characteristic (ROC) curve analysis was used in order to determine retroreflectivity values that have level 2 and 3 quality detection with a sensitivity of 95%.

In all tests, the significant level was set at 0.05. IBM SPSS 26 was used for the statistical analysis.

## 3. Results

The presentation of results is divided into three sections: (1) descriptive statistics, (2) correlation analysis and (3) ROC analysis. 

### 3.1. Descriptive Statistics—Retroreflectivity, Lane Markings’ Detection Quality and Range of View

The average retroreflectivity (R_L_) ranged from 0 to 661 mcd/lx/m^2^ with mean value of 197.50 mcd/lx/m^2^ (SD = 118.83). The Kolmogorov–Smirnov test indicated a statistically significant departure of R_L_ from a normal distribution (z(2150) = 0.078; *p* < 0.001), mostly due to a relatively large number of zero-value readings (12.6%). For this reason, median and interquartile ranges were used as additional measures of central tendency and variability. The median value of R_L_ was 201 mcd/lx/m^2^ with the interquartile range from 109 to 290 mcd/lx/m^2^.

As stated in Section 2.3, the retroreflectivity and the detection quality were categorized. Most of the samples (28.4%) had retroreflectivity between 200–300 mcd/lx/m^2^ followed by 100–200 mcd/lx/m^2^ (27.0%), while almost the same number of samples had R_L_ below 100 mcd/lx/m^2^ (22.6%) or higher than 300 mcd/lx/m^2^ (22.0%). On the other hand, more than half of the measurements had the highest level of detection quality (63.1%), 20% had level “2” quality, while level “1” and “0” had less than 10% each.

The average range of view when road markings were detected by Mobileye ranged from 0 to 74.89 m, and its mean value was 37.79 m (SD = 18.98). The distribution of values related to the range of view, as indicated by the Kolmogorov–Smirnov test (z(2150) = 0.054; *p* < 0.001), was not normally distributed again, mostly due to the number of zero-value readings (5.9%). The median value of the average range of view was 40.28 m, and the interquartile range ranged from 25.45 to 52.0 m.

The summary of descriptive statistics is presented in Table 3.

### 3.2. Correlation Analysis

Due to departures of distributions of the average R_L_ and the average range of view from a normal distribution, as well as the fact that the average detection quality was measured on an ordinal scale, Spearman’s correlation coefficient was used as a measure of association between the aforementioned variables (Table 4). Both the average detection quality and the average range of view were significantly positively correlated with the average R_L_, but the average detection quality was to a larger degree determined by the average R_L_ compared to the average range of view. 

The Kruskal–Wallis test indicated that there is a statistically significant difference in the average range of view between categories of the average R_L_ (χ2 (3) = 329.84, *p* < 0.001); therefore, categories were additionally compared using a series of Mann–Whitney U tests with a Bonferroni-corrected level of statistical significance. The average range of view was significantly lower for the first R_L_ category ranging from 0 to 99 mcd/lx/m^2^, compared to the second category ranging from 100 to 199 mcd/lx/m^2^ (U = 62457.5, z = −15.71, *p* < 0.001), the third category ranging from 200 to 299 mcd/lx/m^2^ (U = 73191.0, z = −14.42, *p* < 0.001) and the fourth category with the highest R_L_ (U = 50998.0, z = −14.92, *p* < 0.001). The second category of R_L_ (ranging from 100 to 199 mcd/lx/m^2^) did not significantly differ in the average range of view compared to the third category ranging from 200 to 299 mcd/lx/m^2^ (U = 167408.0, z = −1.65, *p* = 0.594), nor compared to the fourth category of 300 mcd/lx/m^2^ and higher R_L_ (U = 13149.0, z = −1.11, *p* > 0.999). Finally, no significant difference was found between the third and the fourth category of retroreflectivity (U = 141432.0, z = −0.56, *p* > 0.999). A summary of the aforedescribed results is presented in Table 5.

When looking at the differences in average R_L_ between detection quality categories, the results of the Kruskal–Wallis test indicate that a statistical difference exists (χ2 (2) = 602.77, *p* < 0.001). As in the previous case (differences in the average range of view between categories of average R_L_), a series of Mann–Whitney U tests with a Bonferroni-corrected level of statistical significance were conducted to further explore those differences. In cases where the detection quality was 0 or 1, the average R_L_ was significantly lower compared to the case where the detection quality was 2 (U = 41946.5, z = −11.49, *p* < 0.001) or 3 (U = 62095.0, z = −22.06, *p* < 0.001). Additionally, the average R_L_ in cases of highest detection quality was significantly higher compared to cases where the detection quality was 2 (U = 157430.5, z = −14.29, *p* < 0.001).

### 3.3. ROC Curve Analysis

In order to determine the value of R_L_ that has a minimum of level 2 detection quality with a sensitivity of 95%, ROC curve analysis was employed. The area under the curve, i.e., the probability that R_L_ from a randomly chosen observation with a minimum of level 2 detection quality will be larger than the one from a randomly chosen observation with the detection quality level smaller than 2 was 0.84, *p* < 0.001, 95% CI: 0.82–0.87 (Figure 2). Since it was statistically significant, this indicates that the use of R_L_ values to predict a minimum of level 2 detection quality is better than random guessing.

The value of R_L_ at which the sensitivity was 95% for a minimum of level 2 detection quality was 54.5 mcd/lx/m^2^. The sensitivity or the proportion of observations with level 2 or 3 of detection quality that also had an R_L_ of at least 54.5 mcd/lx/m^2^ in the total number of observations with level 2 or 3 of detection quality was 95% (95% CI: 94–96%). The specificity or the proportion of observations with a detection quality level lower than 2 and R_L_ smaller than 54.5 mcd/lx/m^2^ in the total number of observations with the detection quality level smaller than 2 was 56% (95% CI: 52–60%). The positive predictive value or the probability of level 2 or 3 of detection quality, given that R_L_ is 54.5 mcd/lx/m^2^ or higher, was 91% (95% CI: 91–92%). The negative predictive value or the probability of detection quality level lower than 2, given that R_L_ is lower than 54.5 mcd/lx/m^2^, was 70% (95% CI: 65–74%). Since both positive and negative predictive values depend on the proportion of observations with level 2 or 3 of detection quality in the sample which was 1784/2150 (83%, 95% CI: 81–85%), positive and negative likelihood ratios were computed as more stable measures. The positive likelihood ratio was 2.16 (95% CI: 1.97–2.37), meaning that an observation with an R_L_ value of at least 54.5 mcd/lx/m^2^ would have more than two times larger odds to have a minimum of level 2 detection quality rather than lower than 2. The negative likelihood ratio was 0.09 (95% CI: 0.07–0.11), meaning that an observation with an R_L_ value lower than 54.5 mcd/lx/m^2^ would have about 10 times smaller odds for a minimum of level 2 detection quality rather than lower than 2. The total accuracy, i.e., the overall proportion of correct detection quality classifications based on R_L_ value, was 88% (87–90%). The aforementioned results are presented in Table 6.

The ROC curve analysis was also used to determine the value of R_L_ for level 3 detection quality with a sensitivity of 95%. The area under the curve in Figure 3 was 0.80, and it was statistically significant (*p* < 0.001, 95% CI: 0.78–0.82).

The value of R_L_ at which the sensitivity was 95% for level 3 detection quality was 88.5 mcd/lx/m^2^. The sensitivity or the proportion of observations with level 3 detection quality and R_L_ values of at least 88.5 mcd/lx/m^2^ in the total number of observations with level 3 detection quality was 95% (95% CI: 94–96%). The specificity or the proportion of observations that had the detection quality level lower than 3 and, at the same time, R_L_ lower than 88.5 mcd/lx/m^2^ in the total number of observations with a detection quality level lower than 3 was 45% (95% CI: 43–47%). The probability of level 3 detection quality, given that R_L_ is 88.5 mcd/lx/m^2^ or higher (i.e., positive predictive value), was 75% (95% CI: 74–76%), while the probability that the detection quality level was lower than 3, given that R_L_ is lower than 88.5 mcd/lx/m^2^ (i.e., negative predictive value), was 84% 95% CI: (80–87%), as shown in Table 7. Positive and negative likelihood ratios were also determined. The ratios are not dependent on the proportion of observations with level 3 detection quality in the sample, which was 1356/2150 (63%, 95% CI: 61–65%). The positive likelihood ratio was 1.73 (95% CI: 1.66–1.81), so the observation with an R_L_ value of at least 88.5 mcd/lx/m^2^ would have more than 1.5 times larger odds to have level 3 detection quality rather than lower than 3. The negative likelihood ratio was 0.12 (95% CI: 0.09–0.15); therefore, an observation with an R_L_ value smaller than 88.5 mcd/lx/m^2^ would have about 10 times smaller odds for level 3 detection quality rather than lower than 3. The total accuracy, i.e., the overall proportion of correct detection quality classifications based on R_L_ value, was 77% (95% CI: 75–78%).

## 4. Discussion

Although several studies [12,15,16] investigated how different factors affect lane detection and, thus, proper functioning of LSS, gaps in the literature still exist. Mainly, these gaps are related to determining adequate levels of lane markings’ visibility in different conditions. For this purpose, the aim of this study was to investigate how the retroreflectivity of middle road markings (nighttime visibility) affects the functioning of LSS. 

The results from an on-road investigation indicate that the retroreflection of road markings significantly impacts both the detection quality and the average range of view. In other words, higher retroreflection provided a higher level of detection quality and longer range of view. The range of view of markings with retroreflectivity below 100 mcd/lx/m^2^ was significantly lower compared to all other R_L_ categories with retroreflectivity higher than 100 mcd/lx/m^2^. On the other hand, no statistical difference was found between markings with retroreflectivity higher than 100 mcd/lx/m^2^. The median value of the average range of view was 40.28 m, and the interquartile range ranged from 25.45 to 52.0 m. When looking at the detection quality, the results suggest that, when it was 0 or 1, the average R_L_ was significantly lower compared to the cases where the detection quality was 2 or 3. Furthermore, the average R_L_ for level 3 detection quality was significantly higher compared to cases where the quality was level 2. These results further support findings from previous studies [12,27,28]. Namely, during nighttime, the visibility of markings is achieved with the use of retroreflective materials (glass beads), which return the incoming light rays from the vehicle headlights back to the driver, thus creating contrast between the marking and the dark environment. The higher the retroreflectivity, the higher the contrast ratio between markings and the environment.

The ROC curve analysis indicated that markings with retroreflectivity of at least 54.5 mcd/lx/m^2^ will enable at least level 2 quality reading by machine-vision (with 95% sensitivity). Furthermore, such markings (R_L_ ≥ 54.5 mcd/lx/m^2^) will have more than two times larger odds to have a minimum of level 2 detection quality rather than lower than 2. For level 3, which represents the highest level of detection quality, the markings should have at least 88.5 mcd/lx/m^2^ (with a sensitivity of 95%). Markings that have the aforementioned retroreflectivity will have more than 1.5 times larger odds to have level 3 detection quality rather than lower than 3. These findings further support the findings from off-road tests conducted by Texas A&M Transportation Institute [22] and, to some extent, findings from on-road tests conducted in Sweden [26]. 

Although level 2 detection quality is sufficient to enable the functioning of LSS, meaning that sufficient retroreflectivity should be above 55 mcd/lx/m^2^, it has to be noted that the presented results were obtained from testing conducted in almost ideal conditions (dry weather, controlled driving speed, no impact of glare from the vehicles from the opposite direction, etc.). Of course, conditions on the road may vary, which can ultimately positively or negatively affect the accuracy of machine-vision. Therefore, in order to ensure the proper functioning of LSS in diverse conditions, our results support the recommendation from the literature [12,28] that markings should have at least 100 mcd/lx/m^2^ retroreflectivity at all times. Even though the study was conducted on two-way rural roads, the results are also applicable for motorways. Indeed, motorways mostly have similar geometry along their entire length; they are better maintained compared to rural roads, they have wider road markings and road marking standards for motorways are higher compared to rural roads in most countries. Due to the aforementioned reasons, it is reasonable to expect that the recommended minimal retroreflectivity (100 mcd/lx/m^2^) of road markings should also facilitate the proper functioning of LSS on motorways.

Although this study provided valuable results, there are some limitations. Namely, road geometry was not taken into account. The literature [12] suggests that road geometry may influence the accuracy of machine-vision to some extent. Further research is needed to identify how different road geometries (for example, horizontal and vertical curves, etc.) affect machine-vision and how the visibility of road markings influences their detection in such situations. Furthermore, the configuration of dashed middle lines were not evaluated, although existing literature indicates that solid lines are detected “better” compared to dashed lines with the same characteristics (equal width, brightness and maintenance) [12]. However, due to the lack of data related to the exact location of each type of dashed line, such analysis was not conducted. 

In order to obtain a deeper insight into the necessary minimal levels of retroreflection of road markings for machine-vision, future studies should focus on a detailed investigation of the influence of other factors on LSS, such as weather conditions, road geometry, markings characteristics (colour, width, configuration), driving speed, pavement characteristics (type, condition and texture) etc., as well as different combinations of these factors. Additionally, future studies should test several machine-vision systems in order to obtain a broader picture of how their accuracy may vary, especially depending on the quality of road markings.

## 5. Conclusions

The results of this study confirm that road markings’ retroreflectivity significantly affects the accuracy of machine-vision during nighttime. Namely, with the increase in retroreflectivity, the detection quality and the range of view increase. Based on the ROC analysis, it was determined that the minimal value of retroreflection for a minimum of level 2 detection should be above 55 mcd/lx/m^2^, and 88 mcd/lx/m^2^ for best detection quality (level 3). Therefore, we recommend that road markings should have at least 100 mcd/lx/m^2^ retroreflection at all times in order to enable accurate detection by machine-vision in diverse conditions. The results of this study are valuable to researchers and practitioners from the field and may be useful for developing standards and minimal requirements for road markings to facilitate the proper functioning of ADAS. The minimal retroreflectivity level obtained and recommended in this study may also help road authorities in planning and optimizing their maintenance activities.

## Figures and Tables

**Figure 1 sensors-22-01303-f001:**
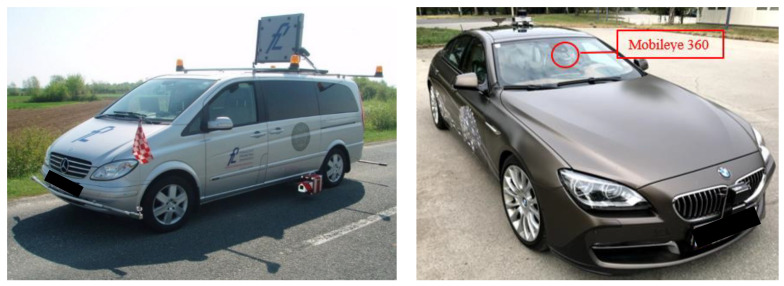
Testing vehicles equipped with measuring devices: dynamic retroreflectometer Zehntner ZDR 6020 (**left**) and Mobileye 360 (**right**).

**Figure 2 sensors-22-01303-f002:**
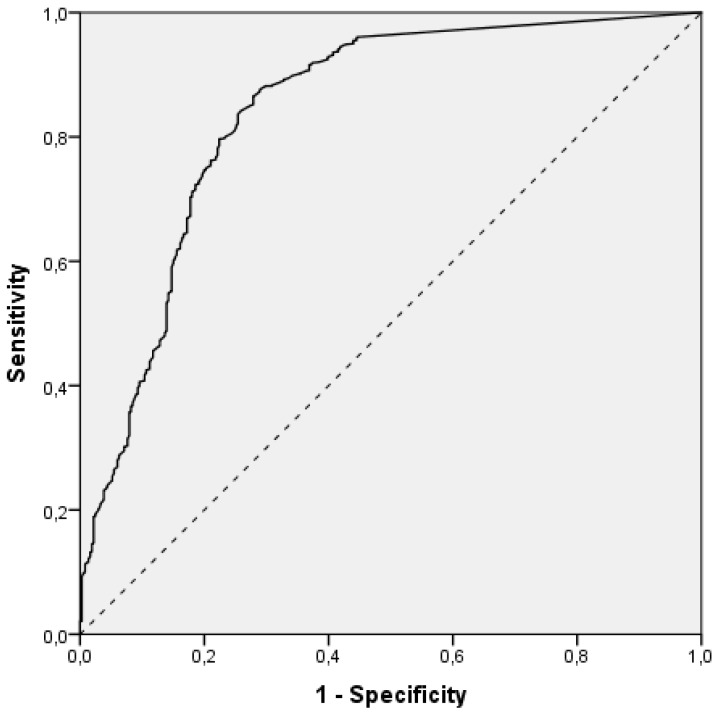
ROC curve of R_L_ for differentiation of detection quality below 2, 2 or more.

**Figure 3 sensors-22-01303-f003:**
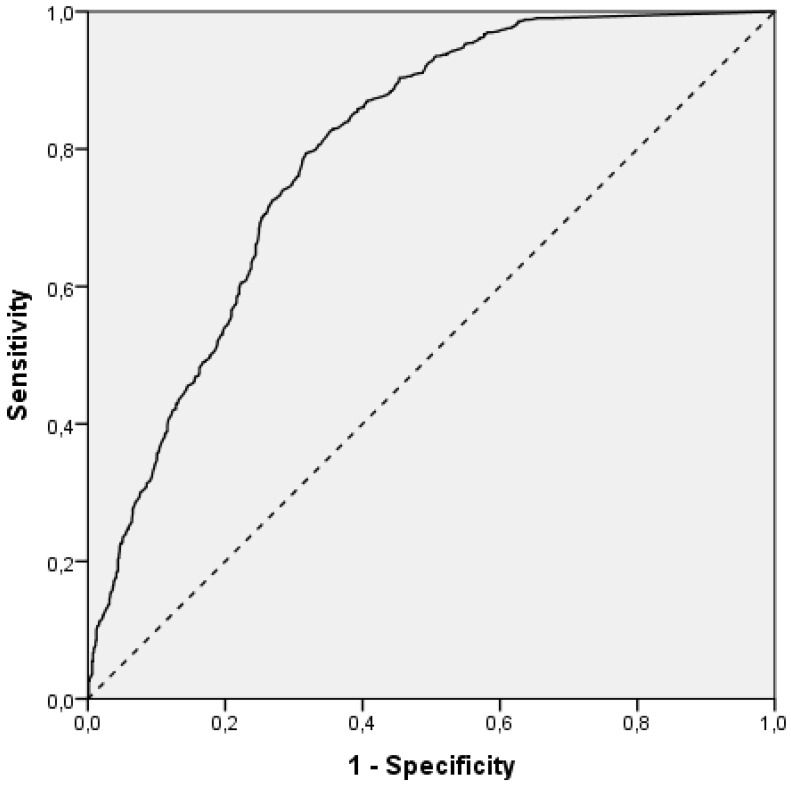
ROC curve of R_L_ for differentiation of detection quality below 3 and 3.

**Table 1 sensors-22-01303-t001:** Basic technical specification of the used Mobileye system.

Vision Sensor	
Vision Sensor:	Aptina MT9V024 (1/3″) RCC
Array Format:	Total: 752 H × 480 V—Active pixels: 640 H × 480 V
Pixel Size:	6.0 μm × 6.0 μm
Dynamic Range:	>55 dB linear; >100 dB in HDR mode
Responsivity:	4.8 V/lux sec (550 nm)
Angle of view:	38° (horizontal)
Focus range:	5 m to infinity
AGC:	Automatic Gain Control of the image sensor for high dynamic range
**EyeQ2 Vision Processor**	
332 MHz clock rate running seven parallel processes	
Two MIPS24KF 32bit CPUs	
Eight 64bit Vision Computing Engines (VCE)	
Eight channels DMA	
64bit width 512 KB on-chip SRAM	

Source: [32].

**Table 2 sensors-22-01303-t002:** Characteristics of tested road sections.

#	Length of the Road (km)	Width of the Markings (cm)	Length of the Middle Line (km)	Age of the Middle Marking
1	32.21	15	Solid: 20.61 Dashed: 11.60	<6 months
2	20.53	15	Solid: 14.68 Dashed: 5.85	<6 months
3	38.05	15	Solid: 15.00 Dashed: 23.05	<6 months
4	30.01	15	Solid: 30.08	>1 year

**Table 3 sensors-22-01303-t003:** Summary of descriptive statistics.

			Range of View (m)	
Average R_L_ (mcd/lx/m^2^)	*n*	(%)	Median	Interquartile Range
<100 mcd/lx/m^2^	486	(22.6)	22.41	0.96–38.14
≥100 < 200 mcd/lx/m^2^	581	(27.0)	44.47	31.68–54.97
≥200 < 300 mcd/lx/m^2^	610	(28.4)	42.68	29.00–54.67
≥300 mcd/lx/m^2^	473	(22.0)	42.94	32.69–52.73
**Average detection quality**				
0	168	(7.8)	0	0–1.60
1	198	(9.2)	20.77	11.56–32.79
2	428	(19.9)	34.81	25.61–44.12
3	1356	(63.1)	46.01	35.70–55.60

**Table 4 sensors-22-01303-t004:** Spearman’s coefficients of correlation between examined variables.

	1. Average R_L_	2. Average Detection Quality
1. Average R_L_	-	
2. Average detection quality	0.53	-
3. Average range of view	0.29	0.52

Note. All coefficients are significant at the *p* < 0.001 level, two-tailed.

**Table 5 sensors-22-01303-t005:** Summary of difference in average range of view between categories of average R_L_.

R_L_ Categories (mcd/lx/m^2^)	1	2	3
1 (<100 mcd/lx/m^2^)	-		
2 (≥100 < 200 mcd/lx/m^2^)	*p* < 0.001	-	
3 (≥200 < 300 mcd/lx/m^2^)	*p* < 0.001	*p* = 0.594	-
4 (≥300 mcd/lx/m^2^)	*p* < 0.001	*p* > 0.999	*p* > 0.999

**Table 6 sensors-22-01303-t006:** Diagnostic value of determined R_L_ cut-off point of 54.5 mcd/lx/m^2^ for a minimum of level 2 detection quality.

		Detection Quality		
		2 or 3	<2	
**R_L_** (mcd/lx/m^2^)	≥54.5	true positive 1695	false positive 161	Positive predictive value 91% (91–92%)
	<54.5	false negative 89	true negative 205	Negative predictive value 70% (65–74%)
		Sensitivity 95% (94–96%)	Specificity 56% (52–60%)	

Note. 95% confidence intervals are shown in brackets.

**Table 7 sensors-22-01303-t007:** Diagnostic value of determined R_L_ cut-off point of 88.5 mcd/lx/m^2^ for level 3 detection quality.

No. of Measurements		Detection Quality		
		3	<3	
**R_L_** (mcd/lx/m^2^)	≥88.5	true positive 1285	false positive 434	Positive predictive value 75% (74–76%)
	<88.5	false negative 71	true negative 360	Negative predictive value 84% (80–87%)
		Sensitivity 95% (94–96%)	Specificity 45% (43–47%)	

Note. 95% confidence intervals are shown in brackets.

## Data Availability

The data presented in this study are available on request from the corresponding author.

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
