# Peer review of "Impact of Road Marking Retroreflectivity on Machine Vision in Dry Conditions: On-Road Test"

_sensors, 2022, doi:10.3390/s22041303_

Round 1

Reviewer 1 Report

The paper is well written. It well describes the purpose of the research, the methodology and the equipment.
The paper, however, is more technical than scientific. It is advisable to give the paper a more scientific accent.

Author Response

Thank reviewer for his comment, however we would like to ask for a more detailed clarification on what does reviewer means by “more scientific accent”? From our point of view, we have presented and identified research gap, goal of our research and the methodology. We have conducted several statistical analysis to answer our research question and proposed practical use of the obtained results. Moreover, we identified limitations and of the study and provided suggestions for further studies.

Reviewer 2 Report

The paper presents the correlation between parameters measured in situ in a survey campaign. No original research aspect is present. The presentation of the results is good, as is the structure of the paper. The study results help broaden the literature in the field, but do not add new knowledge. 

Author Response

From the available literature, one can conclude that night-time visibility (retroreflection) represents to some extent impacts the functioning of LSS. However, literature of this topic is limited, and available studies were mainly conducted on a testing track (except Australian study), on which conditions are vastly different compared to the real road conditions.

Due to the limitations of the available literature, we feel that obtained results not only broaden the literature, but also provide a more accurate findings and thus “new” knowledge which may help in developing standards and minimal requirements for road markings, as well as in planning and optimizing their maintenance activities, all with the aim of facilitating proper functioning of ADAS and thus increasing road safety.

Reviewer 3 Report

See attached - unable to attach for unknown reasons

Author Response

Heartfelt thanks to the reviewer for constructive comments and suggestions. The answers to all the comments are in the attached file.
